# An adapted instrument to assess informed consent comprehension among youth and parents in rural western Kenya: a validation study

Muhammed Olanrewaju Afolabi,[1] Stuart Rennie,[2,3] Denise Dion Hallfors,[4] Tracy Kline,[5] Susannah Zeitz,[4,6] Frederick S Odongo,[7] Nyaguara O Amek,[7] Winnie K Luseno[4]

► We included in the submission a copy of the questionnaire that we would like to be published under supplementary file as an appendix

For numbered affiliations see end of article.

**Correspondence to**
Dr Muhammed Olanrewaju Afolabi; Muhammed.Afolabi@lshtm.ac.uk

## ABSTRACT

**Objective** To adapt and validate a questionnaire originally developed in a research setting for assessment of comprehension of consent information in a different cultural and linguistic research setting.

**Design** The adaptation process involved development and customisation of a questionnaire for each of the three study groups, modelled closely on the previously validated questionnaire. The three adapted draft questionnaires were further reviewed by two bioethicists and the developer of the original questionnaire for face and content validity. The revised questionnaire was subsequently programmed into an audio computerised format, with translations and back translations in three widely spoken languages by the study participants: Luo, Swahili and English.

**Setting** The questionnaire was validated among adolescents, their parents and young adults living in Siaya County, a rural region of western Kenya.

**Participants** Twenty-five-item adapted questionnaires consisting of close-ended, multiple-choice and open-ended questions were administered to 235 participants consisting of 107 adolescents, 92 parents and 36 young adults. Test-retest was conducted 2–4 weeks after first questionnaire administration among 74 adolescents, young adults and parents.

**Outcome measure** Primary outcome measures included ceiling/floor analysis to identify questions with extremes in responses and item-level correlation to determine the test-retest relationships. Given the data format, tetrachoric correlations were conducted for dichotomous items and polychoric correlations for ordinal items. The qualitative validation assessment included face and content validity evaluation of the adapted instrument by technical experts.

**Results** Ceiling/floor analysis showed eight question items for which >80% of one or more groups responded correctly, while for nine questions, including all seven open-ended questions, <20% responded correctly. Majority of the question items had moderate to strong test-retest correlation estimates indicating temporal stability.

**Conclusions** Our study demonstrates that cross-cultural adaptation and validation of an informed consent comprehension questionnaire is feasible. However, further research is needed to develop a tool which can estimate a quantifiable threshold of comprehension thereby serving

## Strengths and limitations of this study

► We conducted a cross-cultural adaptability and validation study of an informed consent comprehension tool developed in two differently diverse linguistic settings
► Item-level test-retest reliability, as well as qualitative methods involving face and content validity, were employed to establish reliability and validity of the adapted tool.
► Relatively small sample size and disparate modes of parental consenting posed a unique challenge in validating a tool across many age groups.
► Our tool did not focus on developing a quantifiable threshold of comprehension below which the consent of participants is invalidated.

as an objective indicator of the need for interventions to improve comprehension.

## INTRODUCTION

Informed consent is a key ethical requirement in clinical research. Universally agreed guidelines highlight four elements of informed consent which normally must be satisfied before proceeding with the conduct of scientific research involving human participants. These elements include decisional competence, disclosure of study information, comprehension and voluntariness.[1–4] Of these elements, comprehension of consent information by a prospective research participant is critical to the quality of a consent procedure as it determines how the participant is empowered to use the information to arrive at an informed decision on whether or not to participate in the study.[5] The informed consent process is typically built on the notion that individuals considering participation have demonstrated satisfactory understanding of the consent information.[6]

However, empirical evidence has shown that research participants frequently do not understand significant aspects of the studies they join, such as the difference between participating in clinical research and receiving medical care, that is, 'therapeutic misconception'.[7] They also demonstrate poor understanding of the concepts of randomisation, research risk and benefits and right of withdrawal.[8–10]

Very few studies have assessed research participant comprehension of consent information in African populations. In a systematic review with meta-analysis of 21 studies conducted across several African countries, comprehension of key concepts of informed consent was poor, with less than half of the study participants demonstrating understanding of research concepts such as randomisation and placebo, and with only 30% being aware of participating in clinical research.[11] Conversely, another systematic review focusing on 103 studies conducted mainly in middle-income and high-income countries over a period of 30 years, showed that more than 70% of participants had good understanding of different domains of informed consent including nature of the study, voluntary participation and rights of withdrawal, while appreciable proportions of the participants demonstrated no therapeutic misconceptions and were aware of the study risks and benefits.[12]

This contrast between the ideals of informed consent and the reality of informed consent in practice is especially marked in settings with high illiteracy rates or mistrust of research institutions, or where signatures are rarely employed for transacting business. Overemphasis on written documents can further aggravate these challenges to effective communication, particularly when participants are asked to understand complex information contained in lengthy informed consent documents written in international languages with unfamiliar terms and concepts.[13]

To ensure participants make meaningful decisions that protect their rights and freedom of choice, researchers in socially and economically disadvantaged communities have been advised to make efforts to help prospective participants attain satisfactory understanding of informed consent.[2] To help achieve this, a context-sensitive tool is required to assess participant comprehension of components of consent information delivered during an informed consent discussion. The tool would help to indicate areas of miscomprehension and could further serve as a platform to develop appropriate interventions to improve the identified areas which participants do not understand.

The development and psychometric evaluation of a Digitised Informed Consent Comprehension Questionnaire (DICCQ) has been reported elsewhere.[14] Briefly, the tool was developed following meticulous identification of domains of informed consent which are poorly understood by research participants in low literacy communities in Africa. Owing to the peculiar challenge of inability to read and comprehend informed consent written in

international languages, the questionnaire was developed into an audio computerised tool in the participants' local languages. The tool was administered to assess the understanding of individuals participating in studies taking place in rural and urban settings of The Gambia, a small West African country characterised with an adult literacy rate of less than 50%.[15] Although the tool was reported to be a reliable and valid measure of informed consent comprehension,[14] we expressed concerns regarding whether the tool would retain its acceptable properties if adapted for use in alternate African settings with diverse cultural and linguistic variations.

Given that empirical assessment of consent comprehension is in its infancy and that instrument development and validation are a lengthy but critical process, we focus on the cultural adaptation and evaluation of the DICCQ among a diverse population of adolescents, young adults and parents in a rural setting in western Kenya, East Africa. The initial validation of the DICCQ has been previously published,[14] and is the basis for the instrument which was modified for relevance and tested among the three age groups in Kenya. This work is part of a study on the effects of HIV test disclosure on adolescent behaviour and well-being to inform guidelines for the ethical conduct of adolescent HIV-related research in sub-Saharan Africa. Along with HIV testing, we are investigating comprehension during the informed consent process among parents and youth. The current paper focuses on the first phase of activities to assess informed consent comprehension. The activities included the adaptation of the DICCQ instrument, which was developed for adults, for use among adolescents and their parents, as well as young adults, content validation, ceiling-floor analysis and a test-retest assessment of the adapted instrument. Results will be used to determine the final format of the adapted instrument.

## The original DICCQ: constructs and validation

As highlighted above, the question items on the DICCQ were generated from basic elements of informed consent obtained from literature on guidelines for contextual development of informed consent tools,[13 16–24] international ethical guidelines[3 25] and operational guidelines from The Gambia's National Ethics Committee.[26] Of these, 15 independent domains of informed consent that were not appropriately understood among study participants in low literacy settings were identified. These domains included voluntary participation, rights of withdrawal, study knowledge, study procedures, study purpose, blinding, confidentiality, compensation, randomisation, autonomy, meaning of giving consent, benefits, risks/adverse effects, therapeutic misconception and placebo.

DICCQ was face-validated by a carefully selected panel of researchers with expertise in research methodology and bioethics in the African context. The panel assessed the tool's readability, clarity of words used, consistency of style and likelihood of target participants being able

to answer the questions. This same expert panel also assessed content validity to establish whether the content of the questionnaire was appropriate and relevant to the context for which it was developed.[27] The tool was revised based on the feedback from these experts. The revised questionnaire was further content-validated by randomly selected research assistants and three independent lay persons to assess clarity and appropriateness of the revised question items and their response options.

Given the lack of acceptable systems of writing in Gambian local languages, the question items were audio recorded in three major local languages by experienced native-speaking linguistic professionals who were also familiar with clinical research concepts. Audio back-translations were done for each language by three independent native speakers and corrections were made in areas where translated versions were not consistent with the English version. A final proof of the audio recordings was conducted by three native-speaking clinical researchers who independently confirmed that the translated versions retained the original meaning of the English version.

The revised questionnaire was developed into an audio computer-assisted self-interview (ACASI) format and referred to as the DICCQ.[14] The tool was administered to 250 participants in two studies taking place concurrently in rural and urban Gambian settings. Half of these participants were recalled in 1–2 weeks after the first administration for a retest. Previously published findings showed that the DICCQ had good psychometric properties with potential as a useful tool for measuring comprehension of informed consent among research participants in low-literacy African settings.[14]

For the present study, we adapted the DICCQ for three groups: minor adolescents (15–17 years), their parents and young adults (18–19 years). Although some questions could be considered generic for research studies (such as voluntary participation, confidentiality and rights of withdrawal), others are specific and required adaptation (such as purpose of the study, benefits and risks). For minor adolescents and their parents, questions related to voluntary participation also required adaptation for comprehension of concepts related to adolescent assent and parental permission. In this paper, we describe our validation methods and results, provide the resulting surveys, and discuss issues related to the assessment of comprehension of study information by participants in rural sub-Saharan settings. We refer to the adapted questionnaire as the Informed Consent Comprehension Assessment (ICCA).

## METHODS
### Validation sample
At the start of the parent study, we invited all consented participants from 10 randomly selected village clusters in one subcounty within Siaya County to respond to the ICCA. The first 235 to agree comprised the ICCA

validation sample. Sample size for validation studies is usually determined with the aim of minimising the standard error (SE) of the correlation coefficient for reliability test. Also, 4–10 subjects per question items are recommended to obtain a sufficient sample size in order to ensure stability of the variance-covariance matrix in factor analysis.[28 29] We used these recommendations to determine our sample size.

The validation sample included minor adolescents (n=107), their parents (n=92) and young adults (n=36). Parents were invited if their adolescent child (or children) was selected for the ICCA Study. More than half of the parents (n=49) who took the ICCA were not consented by staff but rather signed a consent form that their adolescent brought home to them.

### Adaptation and validation procedures
We began our adaptation process by developing an ICCA questionnaire for each of the three groups, modelled closely on the DICCQ. We then customised two questions for minor adolescents about voluntary participation (ie, need for parental permission for participation and adolescents' rights to refuse). For parents, questions were adapted as needed to refer to their child as the main study participant. Finally, questions with study-specific content were developed, using content from Instituitional Review Board (IRB)- approved consent forms. The three draft adapted questionnaires were then reviewed by two bioethicists and the developer of the original DICCQ for face and content validity, based on study protocols and the US federal regulations.[4] Suggestions to clarify language and responses from this expert review were incorporated into the second draft.

The revised questionnaire was then programmed for the ACASI format, with translations and back-translations in three languages (Luo, Swahili and English). Next, we conducted pilot tests of the questionnaires with local Kenyan parent and youth advisory group members[30] to determine whether consent form information and ICCA items were consistent/non-contradictory. After each of the three groups (minor adolescents, young adults and parents) completed the appropriate version of ICCA, we asked participants, individually and in separate focus groups, for their opinions about the consent form, ICCA questions, administration of the ICCA using the ACASI format, and staff assistance (if requested) to type in responses to the open-ended questions. Based on feedback from participants, we revised the wording of one question's response categories, dropped one question and revised the consent form to more clearly describe all aspects covered in ICCA.

Subsequently, we administered ICCA to our validation sample 2–4 weeks after consent and immediately prior to the baseline data collection. Adolescents who consented with the parent-child form took the Adolescent ICCA; those who consented with the young adult form took the Young Adult ICCA; and parents took the Parent ICCA. Following recommended guidelines in validation

studies,[28 29] a subset of the sample, n=74, was retested 1–2 weeks later for test-retest analyses. To make the procedure objective, participant selection for the retest was sequential (every second person), stratified by study site. If one refused, staff continued with the sequence (ie, skipping the next eligible and selecting the following).

### Instrumentation

Each ICCA survey consisted of a set of 25 yes/no, multiple-choice and open-ended questions. Responses to the yes/no and multiple-choice questions were coded 0–1 for incorrect/correct answers, respectively. Responses to the open-ended questions were independently coded from completely incorrect to completely correct (0–4) by a panel of three researchers who discussed their scores and, if different, came to a consensus on a single score per case. Responses were also dichotomised (0–1=incorrect; 2–4=correct) for ceiling/floor analysis. The three survey tools (Adolescent, Young Adult and Parent) were generally similar. However, only seven questions and response options were identical across the three samples. Sixteen additional items were identical for adolescents and young adults. Two items were adolescent-specific, two were young adult-specific and 18 items were parent-specific. In addition to the questions on comprehension of informed consent, the ICCA also included sociodemographic items.

### Ethical considerations

Written informed parent/guardian consent and youth assent was obtained for adolescents younger than 18 years old; individuals who were 18 years or older or emancipated minors provided written informed consent. Participation was voluntary and private.

### Patient and public involvement

To ensure the development of the research questions and outcome measures informed the study participants' priorities, experience and preferences, the adapted questionnaires were translated into the preferred local languages of the study participants. Given the technical complexity involved in designing the study, the study participants were not directly involved in this stage. Nevertheless, parent, professional and adolescent advisory committees reviewed all study plans and provided comments. Also, feedback obtained from pilot participants residing in the study area was used to refine the ICCA instruments. We had a team of dedicated staff that was responsible for the recruitment and conduct of the study; the participants were not involved in these processes. There are no plans to organise a feedback forum where the findings reported in this paper will be disseminated to the study participants and other stakeholders. However, findings from the larger parent study will be disseminated to key stakeholders in the study region, including members of our adult community advisory board and youth advisory board.

### Validation and reliability data analysis

All analyses were conducted using Stata V.13.0 (StataCorp, College Station, Texas, USA). First, we conducted descriptive statistics to determine the magnitude of missing data in each of the ICCA items as well as questions with extremes in responding, that is, to which more than 80% in any one group responded correctly or incorrectly (ceiling/floor analyses). Because high comprehension is desirable for ethical consent, we were particularly interested in questions which fewer than 20% of the sample answered correctly, since this may indicate a problem in wording, format or translation, as well as comprehension.

Second, we conducted test-retest analysis to assess temporal stability of the ICCA questions, that is, whether they were reliable in eliciting the same response at initial presentation (test) and at the second presentation 1–2 weeks later (retest). Item-level correlations were examined to determine the test-retest relationships. Due to the data format, tetrachoric correlations were conducted for dichotomous items and polychoric correlations were conducted for ordinal items (open-ended scores) with the user-created polychoric package.[31] We used the following benchmarks to interpret the correlation coefficients: below 0.5 was considered low, 0.5 to 0.69 was moderate, and 0.7 and higher was strong. We interpreted moderate and strong correlation coefficients as indicating acceptable temporal stability. Post hoc analyses, specifically cross-tabulations of participant responses at test and re-test, were conducted to further explore low correlations and to examine relationships in the data where correlation coefficients could not be obtained.

### RESULTS

Table 1 shows the demographics of the validation sample, including age, gender, religion and the relationship between the adolescent and the person who gave permission for the adolescent to join the study. As can be seen, about 71% of adults who gave permission for adolescent study participation identified as parents.

Descriptive analyses showed that there were no questions with more than 5% missing data. The item with the largest percentage amount of missing responses (4%) was the open-ended study risk question (*Are there any bad things that could happen by taking part in this study? If yes, what are they?*). Ceiling/floor analysis showed eight questions for which >80% of one or more groups responded correctly, while for nine questions, <20% responded correctly (table 2). All seven open-ended questions were among the latter category.

As shown in table 3, the great majority of items, when analysed within groupings of the same wording, had moderate to strong test-retest correlation estimates, despite small sample size, suggesting temporal stability. These included all seven items with identical question and response wording for the entire test-retest sample (n=74); 12 of the 16 items with identical question wording

**Table 1** Demographic characteristics of study participants, Kenya, 2017

| Demographics | Adolescents | Young adults | Parents |
|---|---|---|---|
| Age | | | |
| Median | 16 | 18 | 42 |
| Range | 15–17 | 18–19 | 23–95 |
| IQR | 15–16 | 18–19 | 34–53 |
| Gender | | | |
| Male | 60 (56.1%) | 18 (50%) | 22 (23.9%) |
| Female | 47 (43.9%) | 18 (50%) | 70 (76.1%) |
| Currently enrolled in school: N (%) | 105 (98.1%) | 26 (72.2%) | N/A |
| Highest level of education: N (%) | | | |
| Never gone to school | 0 (0%) | 0 (0%) | 6 (6.5%) |
| Did not complete primary (<Std/Class 8) | 72 (67.3%) | 5 (13.9%) | 37 (40.2%) |
| Completed primary (Std/Class 8) | 10 (9.3%) | 7 (19.4%) | 24 (26.1%) |
| Did not complete secondary (<Form 4) | 25 (23.4%) | 24 (66.7%) | 10 (10.9%) |
| Completed secondary (Form 4) | 0 (0%) | 0 (0%) | 12 (13.0%) |
| College or university | 0 (0%) | 0 (0%) | 3 (3.3%) |
| Attended vocational school: N(%) | 0 (0%) | 2 (5.6%) | 12 (13.0%) |
| Religion: N(%) | | | |
| Roman Catholic | 16 (15.0%) | 4 (11.1%) | 16 (17.4%) |
| Protestant/other Christian | 90 (84.1%) | 31 (86.1%) | 76 (82.6%) |
| Muslim | 0 (0%) | 1 (2.8%) | 0 (0%) |
| No religion | 1 (0.9%) | 0 (0%) | 0 (0%) |
| Attending religious services once/week or more: N (%) | 39 (36.4%) | 20 (55.6%) | 52 (56.5%) |
| Relationship with adolescent: N(%) | | | |
| Parent | N/A | N/A | 65 (70.7%) |
| Other | N/A | N/A | 27 (29.3%) |
| Staff present at consenting: N(%) | N/A | N/A | 43 (46.7%) |

and response options for young adults and adolescents (n=45); one of the two questions specific to adolescents (n=33); and 10 of the 18 questions specific to parents (n=29). Seven items, however, had low correlations, while eight could not be estimated because of small sample sizes and/or near perfect correlation.

Three of the 16 items with identical question/response wording for young adults and adolescents had low correlation coefficients ranging between 0.19 and 0.47. Of these, one was the open-ended item, '*What will you be asked to do as a participant in the study after you receive your HIV test results?*' In cross-tabulation, 34 participants (77%) gave the same response at test and retest while, 6 answered correctly at test and incorrectly at retest. For the item, '*What does it mean when you sign the study consent form?*' 26 (58%) gave the same answer at test and retest, while 3 answered correctly at test and incorrectly at retest. For the item, '*Which describes the main benefit of taking part in the study?*' 34 participants (75%) gave the same answer at both test and retest, while 7 answered incorrectly at test and correctly at retest. Finally, a correlation coefficient could not be obtained for the item '*Will you be told your*

*HIV test results during the study?*' because of a lack of variation at retest, with 41 (91%) and 45 (100%) answering correctly at test and retest, respectively.

Of the two items that were specific to adolescents, one had a low correlation coefficient, '*If your parents want you to join the study, but you do not want to, are you still allowed to refuse?*' For this item, 22 (67%) participants gave the same response at test and retest, while 10 answered incorrectly at test and correctly at retest. Correlations for both items specific to young adults could not be run, but cross-tabulations revealed that all answered the question, '*Have you been told that you can freely decide whether you will take part in this study?*' correctly at both test and retest. For the question, '*How did you decide to join the study?*' 10 (83%) answered correctly at test, while all 12 answered correctly at retest.

Of the 18 items with question wording and/or response options specific to parents, three had low correlation coefficients. For the item '*How did you decide that you and your child would join this study?*' 18 participants (62%) gave the same response at test and retest while 8 (28%) answered correctly at retest only. Similarly, for the item, '*If your child*

**Table 2** Ceiling and floor percentages by response group.

| Questions with more than 80% correct (ceiling) | Adolescents (age 15–17 years; n=107) | Young adults (age 18–19 years; n=36) | Parents (n=92) |
|---|---|---|---|
| T-shirt for participation | 93.5 | 97.2 | 80.4† |
| Study activities for youth | 91.6 | 91.7 | N/A |
| HIV test results disclosure | 94.4 | 94.4 | 90.2 |
| Voluntary withdrawal | N/A | 94.4 | 85.9 |
| Decisions for study participation | N/A | 88.9 | N/A |
| What happens if you stop study participation | N/A | 86.1 | N/A |
| Purpose of conducting study | N/A | 88.9 | N/A |
| Voluntary participation | N/A | 100 | 93.5 |
| **Questions with less than 20% correct (floor)** | **Adolescents (age 15–17 years; n=107)** | **Young adults (age 18–19 years; n=36)** | **Parents (n=92)** |
| Mode of group selection | 19.8 | N/A | 17.4† |
| Study benefits | 16.8 | 16.7 | 19.6† |
| Research purpose (open)‡ | 1.1 | 13.3 | 1.1 |
| Study duration (open)‡ | 13.1 | N/A | 9.8 |
| What is next after HIV test results (open)‡ | 14.0 | N/A | N/A |
| Study HIV test versus clinic HCT (open)‡ | 7.7 | 0 | 2.2 |
| Study risks (open)‡ | 9.3 | N/A | 13.0 |
| Whom to call (open)‡ | 10.5 | 19.4 | 19.6† |
| Study eligibility (open)‡ | N/A | N/A | 7.7 |

*Per cent only shown if ceiling/floor cut-off met.
†Parents who consented without staff present would not have met criterion for ceiling; parents who consented with staff would not meet criterion for floor.
‡(open) denotes open-ended questions, (response range=0–4). These were dichotomised for floor/ceiling analysis: 0=0–1, 1=2–4.
N/A, less than 80% of the sample (by population) got these items correct (upper panel) or incorrect (lower panel).
HCT, HIV Counselling and Testing.

tests positive for HIV, will he or she be offered free treatment?' 18 (62%) gave the same response at test and retest and 10 (35%) answered correctly only at retest. For the item, 'Which describes one of the main risks involved in the study?' 19 (68%) gave the same answer at both time points, while 6 (21%) answered correctly only at retest.

Among the five items for which correlation coefficients could not be obtained, 26 participants (90%) answered consistently at test and retest on the question: 'Have you been told that you can freely decide whether you and your child will take part in this study?' For the item, 'Will you and your child be told the results of his or her HIV test results during the study?' 28 participants (97%) answered consistently. For the open-ended item, 'In your own words, can you tell me what makes you and your child eligible to participate in this study?' 25 participants (92%) answered consistently, and 26 participants (90%) answered consistently on the question: 'How long will your child be involved in the study?' For the open-ended item: 'What will you and your child be asked to do as participants in the study after he/she receives their test results?' 23 participants (79%) answered consistently at test and retest. Finally, with the negative correlation (−1.0) on the item, 'What does it mean when you sign the consent form?' 18 parents were consistent at both time points while 10 went from incorrect at test to correct at retest.

## DISCUSSION

The DICCQ[14] proved to be a useful prototype for adaptation with the Kenyan study. Although the parent study was very different from those for which the DICCQ was developed and included minor adolescents and their parents rather than solely adults, we found the comprehensive domain-linked questions highly useful for adaptation. Given the design of our study, we dropped questions related to clinical trials (blinding and placebo), revised questions related to specific study procedures and populations, and added items specific to assenting adolescents. Examination by bioethicists for face and content validity, as well as piloting with relevant local populations, led to further questionnaire revisions. The exercise also led us to clarify some of the information in the informed consent forms.

Psychometric testing (ceiling/floor) led us to modify the open-ended questions as multiple-choice items (see final ICCA versions in online supplementary appendices). We recognise that open-ended items are ideally the better tool for testing comprehension, since participants can guess multiple-choice answers correctly, thus inflating comprehension levels. Nevertheless, we found that writing down answers in their own words (or even telling staff their answers to write them down) was a

**Table 3** Test-retest correlations for questions common to all and specific to adolescents, young adults and parents (n=74)*

| Question | N | Tetrachoric/ polychoric |
|---|---|---|
| **Common to all** | | |
| Have you been given the name and phone number of the person to contact if you have any questions about the study? | 74 | 0.86 |
| Will you receive a T-shirt for taking part in the study? | 74 | 0.6 |
| How were participants selected into different groups in this study? | 74 | 0.57 |
| In your own words, can you tell me what the purpose of the research study is? (open) | 73 | −0.92 |
| What is the difference between taking part in this study and going to the clinic for voluntary HIV testing? (open) | 72 | 0.87 |
| Are there any bad things that could happen by taking part in this study? If yes, what are they? (open) | 70 | 0.9 |
| If you had a question or concern about the study, who would you call? (open) | 74 | 0.72 |
| **Young adults and adolescents** | | |
| Have you been told you can withdraw from the study at any time? | 45 | 0.75 |
| During the study, will anyone not working with KEMRI or the nearest clinic know about your health information? | 44 | 0.62 |
| At what point can you leave the study? | 45 | 0.94 |
| What does it mean when you sign the study consent form?† | 45 | 0.19 |
| What happens if you decide to stop taking part in the study? | 45 | 0.86 |
| Which of the following describes best why the study is being done? | 45 | 0.51 |
| Which of these activities were you asked to take part in today? | 45 | 0.62 |
| Will you be told your HIV test results during the study?‡ | 45 | N/A |
| Other activities you might be invited to do? | 45 | 0.6 |
| If you test positive for HIV, will you be offered free treatments? | 45 | 0.66 |
| If you are invited to participate in additional interviews for this study, how will you be compensated for your participation? | 45 | 0.73 |
| Which describes one of the main risks involved in the study? | 45 | 0.67 |
| Which describes the main benefit of taking part in the study?† | 45 | 0.26 |
| In your own words, can you tell me what makes you eligible to participate in this study? (open) | 45 | 0.9 |
| How long will you be involved in the study? (open) | 45 | 0.86 |
| What will you be asked to do as a participant in the study after you receive your HIV test results? (open)† | 45 | 0.47 |
| **Adolescents only** | | |
| If you want to join the study, but your parent/guardian does not agree, can you still join the study? | 33 | 0.64 |
| If your parents wants you to join the study, but you do not want to, are you still allowed to refuse?† | 33 | 0.45 |
| **Unique to young adults** | | |
| Have you been told that you can freely decide whether you will take part in this study?‡ | 12 | N/A |
| How did you decide to join the study?‡ | 12 | N/A |

*Post hoc analysis with cross-tabulations were used to further explore the low correlation coefficient.
†A correlation coefficient could not be obtained for this item. Cross-tabulations were used to examine relationships within the data.
‡For complete questions with responses, see online supplementary appendix.
KEMRI, Kenya Medical Research Institute.

difficult and off-putting process, and required staff to parse out whether qualitative answers were partially right or wrong. Finally, test-retest correlations suggested moderate to strong temporal stability for items, despite limitations of small sample size and disparate modes of parental consenting.

Our study contributes to ethical discussions about informed consent in Africa in a number of ways. First, the value of a valid and adaptable tool to test comprehension of informed consent in African contexts should be emphasised and articulated. To improve comprehension, one needs an instrument that can reliably identify areas of substandard understanding. With this in hand, these specific areas can then be targeted for interventions. Simply rereading the entire consent document with the participant may not be enough; one may need instead to focus on certain areas (some perhaps specific to the particular study), ask the prospective participant questions and emphasise these areas in a subsequent revisiting of the consent process. Second, the comprehension tool could be feasible for research with human participants conducted in resource-constrained settings. The DICCQ is a free, open-source tool that researchers can adapt to their particular research context, although adaptation comes with some costs. In addition, one could recommend that the tool be used selectively, that is, in large-scale trials involving significant (greater than minimal) risk—where the stakes for valid informed consent are higher—rather than all studies involving human participants. These trials are also more likely than others to have sufficient human and other resources to absorb the costs of adapting and implementing the tool, and its use may be more easily integrated into standard operating procedures. It should be noted that some assessments and interventions can be relatively simple. In a prior study on adolescent perceptions of health services, we assessed the understanding of consent by asking six key questions, and selectively revisiting the consent process depending on the answers.[32] This enhanced consent process targeted adolescents who planned to participate in HIV-related studies where parental permission had been waived. Thirdly, the development and use of the tool could have implications for the ethical review of research. If such tools are feasible and effective in raising comprehension scores, research ethics committees may recommend (or require) their use in the consent processes of (at least a subset of) research studies.

However, some important challenges regarding the use of comprehension assessment tools in consent remain. As some have noted, if full comprehension were a requirement for valid consent, and valid consent was necessary and sufficient for the ethics of research, all research studies involving human participants would likely be unethical.[33] It would be unreasonable—a form of 'research exceptionalism'[34]—to expect vastly higher levels of consent comprehension in research than in other comparable areas of human life. But how much less than full comprehension is 'good enough' for valid informed consent? When should the results of a comprehension assessment trigger the need for interventions to improve understanding?

It is understandable to want a quantifiable threshold of comprehension below which the consent of participants is invalidated. The threshold would provide an objective indicator of the need for interventions to improve

understanding and also provide a goal for such interventions, that is, the intervention should raise comprehension to or above the accepted threshold. It would clearly be worrying, for example, if the comprehension tool revealed that only 5% of study participants understood that they could leave the study at any time, for any reason. If there was an agreed-upon threshold of (say) 65% for understanding that aspect of informed consent, researchers using the tool would know the magnitude of the problem and what to aim for.

However, questions remain about the attainability of such thresholds. First, such thresholds are likely to be affected by contextual factors. For example, it seems plausible that the threshold for understanding study risks should be higher when the risks are higher, and lower when they are lower. Other contextual factors may include the study population involved, nature of the research question or social value of the potential results. If this is the case, the acceptable threshold of comprehension would be a matter of context-sensitive judgement rather than an objective, quantifiable measure. However, comprehension assessment tools still have utility even if this is the case. Results of assessment can help inform 'all things considered' judgements about whether consent comprehension is adequate, particularly when assessments are fine-grained and focus on specific key elements that participants should know. The tool allows researchers to stipulate and test for adequate levels of comprehension (say, 70%) on crucial aspects of research participation, providing research ethics committees with some confidence that serious attention is being paid to this issue. Where to set these levels is likely to become clearer as the tool is used over time. In addition, interventions to improve baseline understanding retain their value even if objective thresholds of acceptable comprehension currently remain elusive. To use an analogy, tools to assess baseline understanding about HIV are valuable even if it is not entirely clear precisely how much you need to know to be a well-informed, responsible citizen.

Finally, for those concerned about quality of informed consent, it should be noted that informed consent is only one element among others in a suite of protections that should be offered to research participants. Even if comprehension seems less than ideal, a study may be morally acceptable if the research is responsibly designed and conducted in other respects.[35] These considerations notwithstanding, our study results reinforce calls to develop innovative and culturally responsive ways to present research-related information, beyond the standard method of reading consent forms.[30] The impossibility of perfect comprehension, as well as the elusiveness of objective thresholds of acceptable comprehension, should not be the enemy of comprehension assessment or evidence-based efforts to improve consent processes.

The study has a number of limitations. Rigorous psychometric testing was beyond the scope of our study and therefore face validation and expert evaluation were used. Sample size for validation was small, particularly

given the differences in instrumentation for our three populations. Ceiling and floor effects, while extensively limiting the item operational range, provided insight into item functioning and informed modifications needed for the ICCA response options, and the current data were recoded to reflect those needs. Further, for test-re-test, we conducted the first ICCA immediately prior to the actual study procedures, and the second after the participants had experienced these procedures, which likely influenced some of their answers at retest. Some parents were not available to meet with staff for consenting procedures, leading to differences in the opportunity to hear the consent form read aloud and to ask questions of staff.

The paucity of similar African studies on instruments for informed consent comprehension is not surprising, given the cost and highly technical nature of psychometric development and testing of a comprehension instrument. Given the difficulties, we found it exceedingly useful to have a non-proprietary instrument that invited adaptation in other contexts. We also found the adaptation and validation process was helpful in further fine-tuning our instrument and our informed consent document, to make sure that we were fully and clearly communicating the information required for human subject protection. We include the final three documents in the online supplementary appendix in the hope that they will be useful to other researchers.

**Author affiliations**
[1]Department of Clinical Research, London School of Hygiene and Tropical Medicine, London, UK
[2]Department of Social Medicine, University of North Carolina, Chapel Hill, North Carolina, USA
[3]Center for Bioethics, University of North Carolina, Chapel Hill, North Carolina, USA
[4]Pacific Institute for Research and Evaluation (PIRE), Chapel Hill, North Carolina, USA
[5]Department of Social, Statistical and Environmental Sciences, RTI International, Research Triangle Park, North Carolina, USA
[6]Department of Health Behavior, University of North Carolina, Chapel Hill, North Carolina, USA
[7]Department of HIV Implementation Science and Services, Center for Global Health Research, Kenya Medical Research Institute (KEMRI), Kisumu, Kenya

**Acknowledgements** The authors thank all the study participants, including the community and youth advisory board members, and field staff who contributed to this Research. This work was done in collaboration with Kenya Medical Research Institute (KEMRI), Center for Global Health Research, Kisumu.

**Contributors** Design of study: WKL, DDH and MOA; drafting and reviewing questionnaires: DDH, WKL, MOA and SR; acquiring data: WKL, FSO and NOA; analysing data: TK, SZ, DDH, WKL and MOA; writing the manuscript: MOA, SR, DDH, WKL, TK, SZ and NOA.

**Funding** The research reported in this publication was sponsored by the National Institute of Mental Health and National Institute of Allergy and Infectious Diseases of the National Institutes of Health Under Award Number R01MH102125 (Winfred (Winnie) Luseno, Principal Investigator)

**Disclaimer** The content is solely the responsibility of the authors and does not necessarily represent the official views of the National Institutes of Health.

**Competing interests** None declared.

**Patient consent** Obtained.

**Ethics approval** Ethical approval was obtained from the Institutional Review Boards of the Pacific Institute for Research and Evaluation (PIRE), USA (IRBNet ID: 601736, Project Code: 0744), and Kenya Medical Research Institute (KEMRI; SSC Protocol No. 2982).

**Provenance and peer review** Not commissioned; externally peer reviewed.

**Data sharing statement** Exclusive use of the data will be maintained by the Principal Investigator (PI), WKL, until the publication of major outputs. Thereafter, following approvals by the institutional review boards of PIRE and KEMRI, deidentified data will made available to the scientific community through requests made to the PI at wluseno@pire.org.

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
