## [Reviewer comments · BMJ Open]

ARTICLE DETAILS

TITLE (PROVISIONAL)	A validation study of an adapted instrument to assess informed consent comprehension among youth and parents in rural western Kenya
AUTHORS	Afolabi, Muhammed; Rennie, Stuart; Halifors, Denise; Kline, Tracy; Zeitz, Susannah; Odongo, Frederick; Amek, Nyaguara; Luseno, Winnie

VERSION 1 – REVIEW

REVIEWER	Elizabeth Cohn Columbia University, USA
REVIEW RETURNED	29-Jan-2018

GENERAL COMMENTS	Well written article describing the adapting and validating of a consent tool for a rural African population. Solid article with interesting findings. A good addition to this body of work.
--

REVIEWER	R.V. Rikard Michigan State University United States of America
REVIEW RETURNED	23-Mar-2018

GENERAL COMMENTS	Review: Adapting and validating an instrument to assess informed consent comprehension among youth and parents in rural western Kenya Manuscript ID: bmjopen-2018-021613 Journal: BMJ Open The authors report on the validation of the Informed Consent Comprehension Assessment (ICCA) among adolescents, young adults and parents in rural western Kenya, East Africa. The authors adapted the Digitized Informed Consent Comprehension Questionnaire (DICCQ) to determine if the ICCA would retain acceptable psychometric properties given the cultural and logistic variation across Africa. The authors administered the ICCA to a validation sample of 235 respondents of whom 74 were re-tested 1-2 weeks later for the test-retest analyses. The results reveal that while the sample size is small; the test-retest correlations suggest moderate to strong temporal stability for the ICCA items. The authors contend that the research contributes to the need for a valid and adaptable tool to assess comprehension of informed consent in African communities and an assessment for research in resource-constrained environments. The manuscript is well written and employs adequate assessment of the ICCA. The authors acknowledge the limitations of the research; yet, the authors make a compelling argument regarding the research
--

	contribution. The connection between the DICCA and ICCA is clear as well as the necessity for the ICCA. I believe the research is important and recommend acceptance of the manuscript with no revisions.
--	---

REVIEWER	Rebecca Jessup Monash University and Cabrini Institute, Australia
REVIEW RETURNED	29-Mar-2018

GENERAL COMMENTS	Title: Adapting and validating an instrument to assess informed consent comprehension among youth and parents in rural western Kenya This is a really interesting study and addresses an important issue in both health research and health service delivery (where informed consent is required) for low literate communities. The authors should be commended for attempting to address this in their research. The discussion about quantifiable thresholds of comprehension was really interesting and very important given that some of the most pertinent concepts for understanding in consent were not well understood by all age groups in this study. However, I do not feel that the translation and validation processes used in this study met minimum requirements and at this stage I do not feel the paper is at an appropriate standard for publication in the BMJ. Specific feedback has been provided below: Line 122 – 124 Who has expressed concern? If not authors, this statement requires a reference? If authors, you need to state this? Line 129 If this is a modified version of the instrument, how was it modified? How were items chosen for inclusion/ exclusion against the original DICCCQ? Line 130 This study appears to be part of a large study. Was ethics given to conduct this study, and if so, who provided it and what is the study number. Line 182 How was a sample size of 235 decided on? How was the test-retest sample size of 74 (Line 276) decided on or was this a convenience sample and if so, is it representative? Table 2: There are a lot of empty cells in Table 2– what does this mean? Were these items not applied to some populations in your sample? Line 341 – 345 The methods for item reduction and modification outlined in this paragraph should form part of your methods, and would address concerns identified above (Line 129). The authors have not taken enough steps to ensure linguistic and cultural validation (equivalence of language as well as cultural/ conceptual equivalence to ensure the intended meaning of the items is maintained) of the translated instrument. They have misused the term ‘back translation’ in Line 157 and appear to have completed a forward translation only, performed by a single translator for each language, with checks by a single bilingual researcher for each language. A more robust method would involve at least two bilingual researchers for each language conducting linguistic and cultural validation of the tool following translation, before a back translation process where the instrument is translated back from the new language to English and then rechecked for linguistic and cultural validation. The WHO process of translation and adaptation of instruments is a useful guide for this process http://www.who.int/substance_abuse/research_tools/translation/en/ The adapted instrument has not been assessed for validity against the original DICCCQ, and has only undergone a reliability assessment and a single assessment of internal validity using floor/ ceiling
---

	effects. As the instrument is not a direct translation but a modified version targeting a different age group with select item inclusion, it is uncertain as to whether the internal validity of the original instrument has been maintained. To ensure maintenance of this validity, ideally bilingual subjects should have been given both language versions of the instruments at two different times in random order and then some measure of internal consistency (Cronbach's alpha or composite reliability) should have been applied to the dataset. Experts in instrument development and translation should ideally have been involved in this process.
--	---

VERSION 1 – AUTHOR RESPONSE

Reviewers' comments:

Reviewer #1:

Well written article describing the adapting and validating of a consent tool for a rural African population. Solid article with interesting findings. A good addition to this body of work.

Response: We thank the reviewer and appreciate their positive feedback on our manuscript

Reviewer #2:

The authors report on the validation of the Informed Consent Comprehension Assessment (ICCA) among adolescents, young adults and parents in rural western Kenya, East Africa. The authors adapted the Digitized Informed Consent Comprehension Questionnaire (DICCQ) to determine if the ICCA would retain acceptable psychometric properties given the cultural and logistic variation across Africa. The authors administered the ICCA to a validation sample of 235 respondents of whom 74 were re-tested 1-2 weeks later for the test-retest analyses. The results reveal that while the sample size is small; the test-retest correlations suggest moderate to strong temporal stability for the ICCA items. The authors contend that the research contributes to the need for a valid and adaptable tool to assess comprehension of informed consent in African communities and an assessment for research in resource-constrained environments.

The manuscript is well written and employs adequate assessment of the ICCA. The authors acknowledge the limitations of the research; yet, the authors make a compelling argument regarding the research contribution. The connection between the DICCQ and ICCA is clear as well as the necessity for the ICCA. I believe the research is important and recommend acceptance of the manuscript with no revisions.

Response: We acknowledge with thanks the encouraging words from the reviewer on our manuscript

Reviewer #3:

This is a really interesting study and addresses an important issue in both health research and health service delivery (where informed consent is required) for low literate communities. The authors should be commended for attempting to address this in their research. The discussion about quantifiable thresholds of comprehension was really interesting and very important given that some of the most pertinent concepts for understanding in consent were not well understood by all age groups in this study.

However, I do not feel that the translation and validation processes used in this study met minimum requirements and at this stage I do not feel the paper is at an appropriate standard for publication in the BMJ.

Specific feedback has been provided below:

Line 122 – 124: Who has expressed concern? If not authors, this statement requires a reference? If authors, you need to state this?

Response: Following review of the findings published on the reliability of DICCA (Afolabi et al, 2014 BMJ Open), the authors expressed concerns whether the tool would retain these properties, this formed the basis for the current study. This change has been reflected in Line 123 of the revised manuscript.

Line 129: If this is a modified version of the instrument, how was it modified? How were items chosen for inclusion/ exclusion against the original DICCCQ?

Response: A detailed description of the process of modification of DICCCQ to ICCA was given in Lines 171-176 and 195-204.

Line 130: This study appears to be part of a large study. Was ethics given to conduct this study, and if so, who provided it and what is the study number.

Response: Ethical approval was sought and obtained. The detailed information was indicated in Lines 237-238 of the manuscript to which we have added the associated ethics review board protocol numbers.

Line 182: How was a sample size of 235 decided on? How was the test-retest sample size of 74 (Line 276) decided on or was this a convenience sample and if so, is it representative?

Response: We have now clarified how we determined our sample (see lines 185-189 and lines 218-219). Sample size for validation studies is usually determined with the aim of minimising standard error of the correlation coefficient for reliability test. Also, 4-10 subjects per question items are recommended to obtain a sufficient sample size in order to ensure stability of variance-covariance matrix in factor analysis (Kline P, 2000; Nunnally & Bernstein, 1994). Given these recommendations and that ICCA had 25 question items, we decided on 235 as a good sample size for the validation study. Following recommended guidelines in validation studies, re-test analysis was done in a sub-set of the study participants. To make the procedure objective, a sequential stratified method was adopted. This might explain the low response rate but did not negatively impact on our findings. This was also highlighted under the study limitations in lines 442-451

Table 2: There are a lot of empty cells in Table 2– what does this mean? Were these items not applied to some populations in your sample?

Response: These cells were not applicable in that less than 80 percent of the sample (by population) got these items incorrect or correct (depending on the section of the table), and we have reflected these in the revised table

Line 341 – 345: The methods for item reduction and modification outlined in this paragraph should form part of your methods, and would address concerns identified above (Line 129).

Response: We appreciate the reviewer's comments and have reflected the suggestions in Lines 135-138 of the Introduction.

The authors have not taken enough steps to ensure linguistic and cultural validation (equivalence of language as well as cultural/ conceptual equivalence to ensure the intended meaning of the items is maintained) of the translated instrument. They have misused the term 'back translation' in Line 157

and appear to have completed a forward translation only, performed by a single translator for each language, with checks by a single bilingual researcher for each language. A more robust method would involve at least two bilingual researchers for each language conducting linguistic and cultural validation of the tool following translation, before a back-translation process where the instrument is translated back from the new language to English and then rechecked for linguistic and cultural validation. The WHO process of translation and adaptation of instruments is a useful guide for this process http://www.who.int/substance_abuse/research_tools/translation/en/

Response: Concerning back-translations, Lines 157-163 referred to the audio back-translations that were conducted for the original tool DICCCQ to circumvent the challenges posed by lack of standardized written Gambian local languages. The process and reference cited by the reviewer applies to written languages where written forward and back-translations are recommended. These guidelines were also followed in the forward and back-translations of DICCA and ICCA with acceptable modifications.

The adapted instrument has not been assessed for validity against the original DICCCQ, and has only undergone a reliability assessment and a single assessment of internal validity using floor/ ceiling effects. As the instrument is not a direct translation but a modified version targeting a different age group with select item inclusion, it is uncertain as to whether the internal validity of the original instrument has been maintained. To ensure maintenance of this validity, ideally bilingual subjects should have been given both language versions of the instruments at two different times in random order and then some measure of internal consistency (Cronbach’s alpha or composite reliability) should have been applied to the dataset. Experts in instrument development and translation should ideally have been involved in this process.

Our team consists of experts in instrument development and psychometric testing who deployed their wealth of experience in successfully addressing unique challenges posed in validating ICCA in low literacy setting using a digitalized audio channel. Also, we clearly discussed with convincing reasons in Lines 442-451 why conventional psychometric validation was not feasible in the context of assessing the validity of DICCCQ against ICCA. Nevertheless, the qualitative methods we adopted (face validity and expert evaluation) objectively demonstrated the validity and usefulness of ICCA tool.

VERSION 2 – REVIEW

REVIEWER	Rebecca Jessup Monash University and Cabrini Institute
REVIEW RETURNED	07-May-2018
GENERAL COMMENTS	Thank you for providing me with the opportunity to review this manuscript again. I am satisfied that the concerns I had raised have been addressed and am happy to endorse the manuscript for publication in BMJ